

# Validation of the WRF-ARW Eclipse Model with Measurements from the 2019 & 2020 Total Solar Eclipses

Carl E. Spangrude[1], Jennifer W. Fowler[2], W. Graham Moss[3], and June Wang[4,5]

[1]Montana Space Grant Consortium, Montana State University, Bozeman, 59717, USA
[2]NASA Langley Research Center, Hampton, VA, 23666, USA
[3]Department of Physics and Astronomy, University of Montana, Missoula, 59812, USA
[4]New York State Mesonet, State University of New York at Albany, Albany, 12222, USA
[5]Department of Atmospheric and Environmental Sciences, State University of New York at Albany, Albany, 12222, USA

*Correspondence to*: Carl Spangrude (cspangrude@esri.com)

**Abstract.** Field research campaigns in 2019 and 2020 collected hourly atmospheric profiles via radiosonde surrounding the 2 July 2019 and 14 December 2020 total solar eclipses over South America from locations within the paths of eclipse totality. As part of these atmospheric data collection campaigns, the eclipse module of the Advanced Research Weather Research & Forecast (WRF-ARW) model was utilized to model meteorological conditions before, during, and after the eclipse events. The surface and upper air observational datasets collected through these campaigns have enabled further assessment and validation of the WRF-ARW eclipse module's performance in simulating atmospheric responses to total solar eclipses. We provide here descriptions of the field campaigns for both 2019 and 2020 and present results from comparisons of meteorological variables both at the surface and aloft using observational datasets obtained through the campaigns. The paper concludes by recommending further scientific analyses to be explored utilizing the unique datasets presented.

## 1 Introduction

Numerical weather prediction models are utilized widely in the fields of meteorology and climatology for characterizing atmospheric states, general weather forecasting, and research purposes. At present, many such predictive models exist each with various intended applications, inherent assumptions, and levels of complexity. One such numerical model, the Advanced Research Weather Research & Forecast model (WRF-ARW; Skamarock et al., 2008; Skamarock et al., 2021), is of particular interest because it features an eclipse module (WRF-eclipse) for calculating solar eclipses in model simulations. Of course, one major research application of this module is for investigation of atmospheric responses to a total solar eclipse (TSE). WRF-eclipse (Montornès et al., 2016) is implemented here for assessing model performance during two TSE events occurring over South America in 2019 and 2020. Due to their precise temporal and spatial predictability, eclipses can offer valuable research opportunities through measurements of atmospheric responses to the reduction or removal of incoming solar irradiance both at the surface and aloft. However, comprehensive meteorological studies of eclipses are made difficult by the relatively short duration of eclipse totality (i.e., ~2 min) compared to typical model output and observational measurement frequencies. Such measurements (Amiridis et al., 2007; Anderson et al., 1972; Anfossi et al., 2004; Aplin & Harrison, 2003; Eaton et al., 1997; Founda et al., 2007; Giles and Hanna, 2016; Hanna, 2000; Ramchandran et al., 2002;



Szalowski, 2002; Fowler et al., 2019) can be carried out through field research campaigns surrounding an eclipse. Specifically, field campaigns dedicated to high temporal resolution balloon-borne radiosonde observations during a

TSE can produce valuable datasets for establishing a more robust scientific understanding of the atmospheric responses to TSEs, such as eclipse-driven atmospheric gravity waves (AGW). Such campaigns have been performed during the TSEs of 21 August 2017 (Fowler et al., 2019), 2 July 2019 (Colligan et al., 2020), and 14 December 2020, led by the Montana Space Grant Consortium and other collaborators. This study presents basic WRF-eclipse model validation of not only surface variables but also – for the first time – preliminary validation using profile comparisons.


To build upon the findings of eclipse-focused ballooning research campaigns (Fowler et al., 2019), another similar eclipse ballooning campaign was proposed for the 2 July 2019 TSE over South America, described by Colligan et al. (2020). The 2019 campaign collected balloon-borne radiosonde data for 30 h surrounding the 2019 TSE. Soundings collected during the 2019 TSE specifically targeted higher altitudes and were launched with greater temporal

frequency before and after the eclipse, relative to previous campaigns, to maximize stratospheric measurements. A brief overview of the 2019 campaign is provided in Sect. 2. To verify the findings of the 2019 eclipse campaign, a subsequent field campaign was proposed and carried out during the only TSE of 2020 over Chile and Argentina. Figure 1 shows the paths of eclipse totality for the 2019 and 2020 TSEs. These two TSEs over South America presented a unique opportunity to conduct this research; the passage of two TSEs over the same general region and

traveling the same direction (west to east) will not occur again until the 2037 and 2038 eclipses over Australia and New Zealand. In contrast to their relative geographic similarities, the 2019 and 2020 TSEs occurred at different times of day, during opposite seasons in the Southern Hemisphere, and during markedly different meteorological conditions. Further, measurements were performed from a much higher elevation in 2019 relative to 2020. While these differences slightly reduce the feasibility of making direct comparisons between the two TSEs, a general assessment of WRF-

eclipse's ability to simulate eclipse impacts across various meteorological, temporal, and seasonal conditions is thus enabled. If profile data generated by WRF-eclipse are within a reasonable range of accuracy relative to observations from 2019 and 2020, this finding may support the use of model forecasts to estimate future TSE impacts on the atmosphere.

The only TSE of 2020 occurred amid a global pandemic on 14 December over southern South America. Overcoming numerous logistic challenges, a field project entitled *Atmospheric Gravity Wave Radiosonde Field Campaign for Eclipse 2020* was successfully conducted in Chile, collecting surface measurements and hourly radiosonde profiles from two locations 24 hr before, during, and 24 hr after eclipse totality. Prior to the COVID-19 pandemic, the 2020 campaign was intended to complete measurements from four sites spanning the continental extent of the path of

totality. However, due to the increased logistical complexity presented by the pandemic, the final launch locations were ultimately reduced to two locations in Chile. Both launch sites were located along the centerline of eclipse totality, allowing for additional validation of AGW detection methods compared to measurements from a single site as performed by Colligan et al. (2020). The goals of the 2020 campaign were to replicate the methods employed during previous field campaigns (Colligan et al., 2020; Fowler et al., 2019) to study the atmospheric impacts of a TSE and



detect eclipse-induced AGWs in the stratosphere via balloon-borne radiosonde measurements. While totality occurred during the last few hours of daylight for the 2019 TSE, totality of the 2020 TSE occurred at mid-day. Additionally, thick cloud cover and persistent precipitation during the 2020 campaign obscured views of the eclipse and reduced the overall reduction in solar irradiance at the surface caused by the TSE. These meteorological conditions, in stark contrast to the clear skies experienced during the 2019 TSE campaign, offered further opportunity to assess the

performance of WRF-eclipse (Montornès et al., 2016) in simulating TSE events across a range of meteorological conditions. A further description of the 2020 campaign is provided in Sect. 2.2.

The two primary goals of this paper are to: (1) provide descriptions and present comparisons of the field campaigns for the 2019 and 2020 TSEs, and (2) present preliminary results for assessment and validation of the performance of

WRF-eclipse Version 4.3.3 (Skamarock et al., 2021) in simulating atmospheric responses to the 2019 and 2020 TSEs. Though the ability to represent TSEs with WRF-eclipse was included in the fourth major release of WRF-ARW (Skamarock et al., 2019), few studies (Montornès et al., 2016; Moss et al., 2020, 2021; Spangrude et al., 2019) have utilized radiosonde sounding data to validate WRF-eclipse's upper air performance. Observational data obtained during the 2019 and 2020 eclipse field campaigns by both radiosondes and in situ surface weather stations (Spangrude

et al., 2023) are analyzed here to assess and validate the performance of WRF-eclipse in simulating TSEs across a range of environmental and temporal conditions. Section 2 provides descriptions of the 2019 and 2020 eclipse campaigns and describes the modeling methods implemented in this study related to WRF-ARW. Section 3 presents preliminary results from comparisons between measurements and model results for the 2019 and 2020 eclipse events, and the study's resulting conclusions and implications for future research are provided in Sect. 4.

**2 Field Campaigns, Measurements, and Modeling Methods**

**2.1 Description of the 2019 Radiosonde Eclipse Campaign**

The 2019 campaign utilized balloon-borne radiosondes launched hourly before, during, and after eclispe totality to detect eclipse-driven AGWs in the stratosphere. Both surface and profile obervations were made from a single site

within the path of eclipse totality beginning 24 h prior to first contact of the eclipse and continuing until ~3 h after fourth contact. An additional radiosonde flight was performed 30 min prior to totality. First contact (C1) of the 2019 eclipse occurred at 19:22:41 UTC and fourth contact (C4) occurred at 21:46:24 UTC. Eclipse totality occurred at 20:39:17 UTC and lasted approximately 1 min. The radiosonde launch location was determined by intended "coincident measurements with the Andes LiDAR Observatory" (Colligan et al., 2020). By characterizing and

distinguishing wave parameters of AGWs generated by specific sources (e.g., topography, windshear, convection), the 2019 campaign resulted in the first unambiguous detection of eclipse-induced AGWs in the stratosphere (Colligan et al., 2020). The AGW analysis for the TSE of 2 July 2019 is described in detail by Colligan et al. (2020).

The 2019 field site is located in the Coquimbo region, approximately 2 km southeast of Andacollo, Chile, at the

Collowara Tourism Observatory (-30.250° S, -71.063° W; 1283 m). The site has a primarily easterly aspect and little




surface vegetation. During the campaign, 2 m air temperature ranged from 11–21 °C and 2 m winds were generally
low and from the west. The sky was nearly cloud-free for much of the campaign, with "little convective activity" and
"no frontal system passages" (Colligan et al., 2020).

The 2019 campaign personnel were highly trained in radiosonde operations and followed a detailed launch schedule
and flight checklists during the campaign. In total, 25 radiosonde flights were performed hourly between 19:18:00
UTC on 1 July 2019 and 22:35:00 UTC on 2 July 2019. Graw brand DFM-09 radiosondes were suspended from 600
g latex balloons and measured air temperature, pressure, relative humidity, wind speed, and wind direction every 1 s
(~5 m vertically). All radiosondes were initialized using 2 m surface meteorological values provided by an in situ

Lufft brand WS502-UMB Smart Weather Sensor measuring temperature, relative humidity, pressure, wind speed,
wind direction, and solar irradiance every 2 s. Measurement uncertainties for the DFM-09 radiosonde and the WS502-
UMB station are provided by the manufacture technical datasheets. Each radiosonde flight lasted approximately 2 h.
Graw brand GRAWMET software Version 5.12 was used for all ground station systems. The lowest maximum
radiosonde altitude was 10.6 km and the highest maximum altitude was 34.2 km. The average maximum altitude for

the 25 flights was 30.5 km.

WRF-ARW Version 3.8 (Skamarock et al., 2008) was utilized in the planning phase of the 2019 campaign.
Meteorological predictions for the campaign were generated using WRF-ARW for two reasons. Firstly, to provide
local weather predictions for operational and logistical planning. Secondly, to provide Chilean aviation authorities and

regional air traffic controllers with predicted volumes of operational intent for the campaign's radiosonde flight
trajectories, based on local weather conditions surrounding measurements of the TSE. Data collected during the 2019
campaign were subsequently used for preliminiary validation of the WRF-eclipse model by Spangrude et al. (2019).

A major update to the WRF-ARW model from Version 3.9 to Version 4.0 occurred on 8 June 2018. However, WRF-

eclipse was not fully included in Version 4 until the Version 4.3.1 release on 28 October 2021. As such, WRF-ARW
was originally run using Version 3.8, which did include the eclipse module, for the 2019 campaign. Since the inclusion
of WRF-eclipse in the WRF-ARW Version 4.3.1 update, the 2019 WRF-ARW model simulations have been re-run
for comparison to the model simulations of the 2020 TSE, with results presented in Sect. 3. For full clarity, all WRF-
ARW and WRF-eclipse model results presented here were generated with Version 4.3.3 of the WRF-ARW model

(Skamarock et al., 2021).

**2.2 Description of the Atmospheric Gravity Wave Radiosonde Field Campaign for Eclipse 2020**

The 2020 TSE passed over southern South America on 14 December 2020. During the 2020 campaign, surface and

upper air data were collected over two days (13–15 December 2020) from two sites in Chile, Toltén (-39.236° S, -
73.160° W; 3 m) and Villarrica (-39.307° S, -72.104° W; 276 m) (Fig. 1). C1 of the eclipse occurred at 14:39:17 UTC
(14:41:21 UTC) at Toltén (Villarrica) and C4 occurred at 17:28:59 UTC (17:31:22 UTC), respectively. Eclipse totality
lasted approximately 2 m, occurring at 16:01:45 UTC at Toltén and 16:04:12 UTC at Villarrica. Both sites are located



in the Araucanía region in southern Chile, as opposed to the 2019 campaign's location in the Coquimbo region further
to the north. Toltén is less than 10 km from the Pacific coast while Villarrica is located along the Andean foothills,
about 100 km inland from Toltén. The area surrounding Toltén features primarily agricultural and native grasslands
and tributaries into the South Pacific Ocean. At a higher elevation, Villarrica's surrounding environment is relatively
more forested with both deciduous and conifer species. Surface winds at each site were generally low and from the
west during the campaign, though gusts were higher at Toltén due to its coastal proximity. Thick clouds and heavy
rain persisted throughout much of the campaign at each site, however measurements of precipitation were not
performed at Villarrica or Toltén.

During the 2020 campaign, hourly atmospheric soundings were collected from Toltén and Villarrica using helium-
filled 600 g balloons carrying Graw DFM-17 radiosondes. Each radiosonde sounding recorded temperature, pressure,
relative humidity, wind speed, and wind direction every 1 s (~5 m vertically) from the surface into the mid-
stratosphere. Measurement uncertainties associated with the DFM-17 radiosonde sensors are provided by the
radiosonde manufacture's technical datasheet. Each radiosonde ascent lasted approximately 2 h. At each site,
observations of temperature, relative humidity, pressure, wind speed, wind direction, and solar irradiance were
recorded every 2 s at 2 m above ground using Lufft brand WS502-UMB Smart Weather Sensors identical to the 2019
station cited in Sect. 2.1. Surface measurements from the WS502-UMB station were used for initialization of all
radiosondes. While identical surface stations were used for the 2019 and 2020 campaigns, the DFM-17 radiosondes
used in 2020 featured slightly improved measurement uncertainties for wind speed relative to the DFM-09 (0.1 and
0.2 m s$^{-1}$, respectively). Additionally, the DFM-09 calculates pressure using a single GPS while the DFM-17 utilizes
multi-GNSS, resulting in improved uncertainties in pressure, especially aloft. Graw brand GRAWMET software
Version 5.16 was used for all ground station systems.

All radiosonde soundings were collected hourly beginning at 16:01:00 UTC on 13 December 2020 and ending at
18:14:00 UTC on 15 December 2020. Fifty total balloon-borne radiosondes were released hourly from each site
simultaneously. Additionally, a radiosonde was released from each site at a 30 min interval during eclipse totality,
resulting in six active soundings at various altitudes across two locations during maximum totality. Continuous surface
and profile measurements concluded after 50 h, at 18:14:00 UTC on 15 December 2020, corresponding with
termination of the final sounding at each site. Standard operating procedures utilized during the 2019 campaign
(Colligan et al., 2020) were adapted for scaling to multiple teams in 2020.

In 2020, the lowest maximum radiosonde altitude for soundings collected at Toltén (Villarrica) was 16.9 km (10.6
km) and the highest maximum altitude was 36.5 km (36.8 km). The average maximum altitude for soundings collected
at Toltén (Villarrica) was 31.2 km (29.6 km). The maximum average altitude at Villarrica was lower than at Toltén
due to winds aloft and increased interference from topography.



Data quality was maintained throughout both campaigns through extensive team training to ensure adherence to operational standards defined by the *Federal Meteorological Handbook No. 3* (NOAA, 1997). Once all data were collected, three types of screening methods (Meek & Hatfield, 1994) were employed for quality control of the data. These methods include checks for (1) range limits to confirm values are within a reasonably expected range, (2) rate-of-change limits to identify abrupt data changes during flight, and (3) continuity of the data over time to identify

periods of interference when ground receivers were not properly receiving data (Meek & Hatfield, 1994). As a result of careful data quality control and consistent field procedures, 94 of 100 soundings were of high enough quality and completeness to be analyzed (49 from Toltén and 45 from Villarrica) from 2020 and 22 of 25 soundings met the criteria for quality and completeness in 2019. The acceptable soundings from both campaigns each reached the minimally acceptable pressure level of 400 hPa as defined by the NOAA National Weather Service (NWS, n.d.). The

omitted profiles either lost telemetry completely in flight, failed to properly save data files, or did not reach a high enough altitude to be considered in analyses. This study compares the aforementioned surface and profile measurements to model simulations generated by WRF-ARW Version 4.3.3. A further description of the WRF-ARW model is provided in Sect. 2.3.

**2.3 WRF-ARW Models**

WRF-ARW Version 4.2.1 was originally used to generate meteorological forecasts for balloon trajectory prediction during the 2020 campaign. All 2019 and 2020 simulations have subsequently been re-run using WRF-ARW Version 4.3.3 (Skamarock et al., 2021) for validation of WRF-eclipse. A three-domain model simulation with 50 vertical levels

and 250 x 250 grid points (in each domain) was initialized using boundary conditions provided by Global Forecast System (Version 16) data with 0.25° x 0.25° horizontal and hourly temporal resolutions for 2019 and 2020. Models were initialized using the Thompson microphysics scheme (Thompson et al., 2008), Yonsei University Scheme for PBL physics (Hong, Noh, & Dudhia, 2006), and Rapid Radiative Transfer Model for GCMs (RRTMG; Iacono et al., 2008). A 1 m radiation timestep and 30 s integration timestep were used. The model top was set to the 50 hPa pressure

level. The 2020 simulations were configured to begin at 06:00:00 UTC on 13 December 2020 and run through 18:00:00 UTC on 15 December 2020. The 2019 simulations were run from 06:00:00 UTC on 2 July 2019 to 00:00:00 UTC on 3 July 2019. Three geographic model domains were successively nested with a 1:3 parent grid ratio to enable 1 km horizontal resolution in the innermost domain, centered over the launch site(s). Further specifications for WRF-ARW model initialization and configuration are outlined in the supplemental tables S1–S4. Inclusion of TSE

calculations is the only difference in initialization of the WRF-ARW and WRF-eclipse simulations.

Three meteorological variables are examined here for comparison between observations and WRF-ARW/WRF-eclipse model results for the 2019 and 2020 TSE campaigns. Observational data are comprised of surface-based in situ measurements of air temperature (°C) and irradiance (W m$^{-2}$) at 2 m, wind magnitude (m s$^{-1}$) at 2 m (surface

station) and 10 m (radiosonde), and air temperature (°C) profiles up to ~30 km from the 94 soundings collected. Radiosonde soundings are not representative of a vertically continuous column of the atmosphere, but rather can



experience significant horizontal drift during ascent. As such, model grid cells corresponding with both the horizontal and vertical flight paths of soundings are considered in comparisons between model results and the observations presented in Sect. 3.

**3 Results**

**3.1 Comparison of Measured and Modeled Near-surface Variables**

Presented here are near-surface observations of irradiance (W m$^{-2}$), wind speed (m s$^{-1}$), and air temperature (°C) compared against WRF-ARW and WRF-eclipse model results for the 2019 and 2020 eclipses. Two indicators used to

evaluate accuracy of the WRF-ARW models were root mean square error (RMSE) and mean absolute error (MAE) with values from the radiosonde profiles being used as the reference values (Sun et al., 2023). The corresponding equations are:

$$RMSE = \sqrt{\frac{\sum_{i=1}^{N}(T_m - T_r)^2}{N}}$$

$$MAE = \frac{1}{N}\sum_{i=1}^{N}|T_m - T_r|$$

where $T_r$, $T_m$ represent temperature from the radiosonde and temperature from the model respectively, with $I_s$, $I_m$, $w_r$ , and $w_m$ substituting for temperature in evaluating the additional parameters. As a note, results for WRF-ARW in Figures 2–4 are masked by WRF-eclipse results until the time of the eclipse, however both models were run for the entirety of the timeseries. As seen in Figure 2, in situ measurements for the 2019 eclipse campaign concluded prior to conclusion of the 2020 model simulations hence the difference in the timeseries. Figure 2 shows measured irradiance

during the 2020 eclipse was far more dynamic relative to the smoother time series for 2019 (Fig. 2a). Observations of irradiance at 2 m during the 2019 eclipse show an immediate decrease beginning at C1 leading to 0 W m$^{-2}$ during totality (Fig. 2a). WRF-eclipse simulates this decrease in incoming shortwave radiation well, showing close agreement with observations (Fig. 2a) based on a RMSE of 14.9 W m$^{-2}$ and MAE of 13.1 W m$^{-2}$. As expected, results from WRF-ARW do not include an eclipse-related decrease in irradiance hence a RMSE of 107.0 W m$^{-2}$ and MAE of 75.0 W m$^{-2}$

245 .

Both WRF-ARW and WRF-eclipse suppress irradiance too strongly prior to the 2020 eclipse at Toltén and Villarrica, with a RMSE for WRF-eclipse of 75.6 W m$^{-2}$ (49.7 W m$^{-2}$) for Toltén (Villarrica) and a MAE of 36.2 W m$^{-2}$ (26.2 W m$^{-2}$) for Toltén (Villarrica). However, the reduction in irradiance to 0 W m$^{-2}$ during eclipse totality is captured by

WRF-eclipse at both sites (Fig. 2b–2c). After C4, both WRF-ARW and WRF-eclipse results show increased fluctuations in irradiance but do not capture the timing of the fluctuations as compared to observations (Fig. 2b–2c), with a RMSE for WRF-eclipse of 237.4 W m$^{-2}$ (415.7 W m$^{-2}$) for Toltén (Villarrica) and a MAE of 194.4 W m$^{-2}$ (321.1 W m$^{-2}$) for Toltén (Villarrica). For WRF-ARW, the RMSE post-eclipse is 302.0 W m$^{-2}$ (253.9 W m$^{-2}$) for Toltén (Villarrica) and a MAE of 239.4 W m$^{-2}$ (204.5 W m$^{-2}$) for Toltén (Villarrica). Notably, post-eclipse differences in



simulated irradiance between WRF-ARW and WRF-eclipse are resolved relatively quickly, with the models converging 4–6 h after initial divergence (Fig. 2b–2c).

Figure 3 shows comparisons between observations of air temperature (°C) and results from WRF-ARW and WRF-eclipse at 2 m for the 2019 and 2020 eclipses. Both WRF-ARW and WRF-eclipse underestimated temperature by ≥3

°C relative to observations from 2019. WRF-eclipse results for 2019 show a larger decrease in temperature relative to observations during the eclipse (Fig. 3a) with a RMSE for WRF-eclipse of 4.1 °C and a MAE of 3.8 °C. Post-eclipse comparisons between observed and modeled air temperature cannot be made since 2019 measurements concluded at 22:30:00 UTC.

For the 2020 eclipse, WRF-ARW and WRF-eclipse produce identical results in temperature prior to the TSE and agree well with observations at both Toltén and Villarrica (Fig. 3b–3c) with a RMSE for WRF-eclipse of 0.8 °C (0.3 °C) for Toltén (Villarrica) and a MAE of 0.7 °C (0.3 °C) for Toltén (Villarrica). At Toltén, WRF-eclipse diverges from WRF-ARW somewhat after C1, briefly showing a decrease in temperature in agreement with trends in observations (Fig. 3b). Observations from Villarrica do not indicate a significant temperature decrease during the eclipse, however

both models slightly underestimate temperature relative to observations from C1 to totality (Fig. 3c). From C1 to C4 both models perform relatively equally with a RMSE for WRF-eclipse after C1 of 0.6 °C (0.6 °C) for Toltén (Villarrica) and a MAE of 0.5 °C (0.6 °C) for Toltén (Villarrica); for WRF-ARW, the RMSE after C1 is 0.8 °C (0.8 °C) for Toltén (Villarrica) and a MAE of 0.7 °C (0.7 °C) for Toltén (Villarrica).

Figure 4 shows wind magnitude observations at 2 m (surface station) and 10 m (radiosonde) and model results from WRF-ARW and WRF-eclipse at 10 m for the 2019 and 2020 eclipses. Both models overestimate wind magnitude relative to 2019 and 2020 observations. Observations of wind magnitude in 2019 indicate an increase of 1.9 m s$^{-1}$ leading up to C1 and a decrease of 1.25 m s$^{-1}$ during totality with a RMSE of 2.2 m s$^{-1}$ and MAE of 1.9 m s$^{-1}$ prior to C1 and a RMSE of 1.4 m s$^{-1}$ and MAE of 1.2 m s$^{-1}$ during the eclipse (Fig. 4a). WRF-eclipse results diverge from

WRF-ARW during the 2019 eclipse, showing a slight decrease in wind magnitude at totality (Fig. 4a). For the 2020 eclipse at Toltén, results from both models show a decrease in wind magnitude beginning at C1 not seen in the 10 m observations (Fig. 4b) and of a greater magnitude than the 2 m observations. The 2.25 m s$^{-1}$ decrease in wind magnitude from C1 to totality calculated by WRF-eclipse at Toltén is greater than the decrease shown by WRF-ARW (Fig. 4b). Observations at 2 m show a 1.25 m s$^{-1}$ decrease in wind magnitude during totality at Toltén (Fig. 5b). At Villarrica,

both WRF-ARW and WRF-eclipse calculate a decrease in wind magnitude of 2.1 m s$^{-1}$ from C1 to totality which generally aligns with the observational trend (Fig. 4c). Both models show a 6.25 m s$^{-1}$ increase in wind magnitude from totality to C4 while observations suggest a much smaller increase of 1.25 m s$^{-1}$ from totality to just after C4 at Villarrica (Fig. 4c). RMSE for WRF-eclipse from C1 to C4 is 3.9 m s$^{-1}$ (3.1 m s$^{-1}$) for Toltén (Villarrica) and a MAE of 3.7 m s$^{-1}$ (2.3 m s$^{-1}$) for Toltén (Villarrica); the equivalent RMSE calculation for WRF-ARW is 3.4 m s$^{-1}$ (1.7 m s$^{-1}$) for Toltén (Villarrica) and a MAE of 3.1 m s$^{-1}$ (1.4 m s$^{-1}$) for Toltén (Villarrica).




### 3.2 Comparison of Measured and Modeled Profiles

Differences between radiosonde wind magnitude (m s⁻¹) and air temperature (°C) profiles and WRF-eclipse model results at various pressure levels before, during, and after the eclipses of 2019 and 2020 are presented (Fig. 5–6). Measurements at a given pressure level are subtracted from the value at the closest corresponding WRF-eclipse model grid position. Horizontal and vertical model grids are indexed to account for balloon drift during ascent. For 2020 results, pressure levels at (hPa) 980, 750, 500, 250, and 100 are considered. For 2019 results, the bottom-most pressure

level considered is 850 (hPa) since the field site at Andacollo, Chile has a surface elevation of 1283 m.

For 2019, the largest differences between observed air temperature and WRF-eclipse results occurred near the surface. Figure 5a shows WRF-eclipse underestimated air temperature at 850 hPa by 2.5 °C to 4.5 °C before, during, and after the eclipse. In contrast, differences between temperature observations and model results for the 2020 TSE at Toltén

and Villarrica are ≤1.5 °C before and during the eclipse at the 980 hPa and 750 hPa pressure levels (Fig. 5b–5c). The largest differences between observations and model results for both sites in 2020 occurred before and during the eclipse at the 100 hPa pressure level (Fig. 5b–5c). At Toltén and Villarrica, WRF-eclipse overestimated temperature at 100 hPa by 1.75 °C to 4.25 °C before and during the eclipse (Fig. 5b–5c). All three simulations show differences <2 °C before, during, and after the eclipse at the 250 hPa pressure level (Fig. 5a–5c).


Figure 6 shows differences between measured and modeled wind magnitudes before, during, and after the 2019 and 2020 eclipses. In all three cases, the largest differences between measurements and model results occurred at the 250 and 100 hPa pressure levels. Notably, WRF-eclipse greatly underestimated wind magnitude at 100 hPa for Villarrica (Fig. 6c). Table 1 presents the RMSE and MAE values for these measurements.

### 4 Conclusions

The variability of WRF-eclipse's performance in simulating atmospheric responses surrounding a TSE under differing meteorological conditions is made clear through the results presented. Figure 2a highlights the consistency of observed and modeled irradiance under stable, clear-sky conditions. Under such conditions, WRF-eclipse simulates irradiance with relatively high accuracy based on RMSE and MAE values and demonstrates a marked reduction during the

eclipse corresponding well with observations. In contrast, thick cloud cover and heavy precipitation over the 2020 campaign's field sites are most likely responsible for the highly dynamic observed and simulated irradiance values at each site (Fig. 2b–2c), also reflected in the RMSE and MAE values. Both models likely overestimated cloud cover leading up to the eclipse, causing an underestimation of irradiance from 12:00:00 UTC to 16:00:00 UTC on 14 December 2020 (Fig. 2b–2c).


Conversely, both models simulated air temperature with reasonably high accuracy relative to 2020 observations (Fig. 3b–3c), while both underestimated temperature by ≥3 °C relative to observations from the 2019 campaign (Fig. 3a),



which is consistent with the resulting RMSE and MAE values from both campaigns. This potential temperature bias for the 2019 data was shown by Spangrude et al. (2019), however the present results indicate a reduction in this bias

between 15:00:00 UTC and 22:00:00 UTC was achieved, likely through increased resolution of meteorological input data and a lower radiation timestep. Other factors possibly contributing to the variable accuracy of modeled air temperature between the 2019 and 2020 TSEs are the 1000 m difference in elevation between the 2019 and 2020 field sites and significant differences in the surrounding surface vegetation for each campaign.

While results from previous studies on changes in wind magnitude during a TSE are varied, many have shown winds decreasing during TSEs (Fernández et al., 1993, 1996; Ramchandran et al., 2002; Krishnan et al., 2004; Stoev et al., 2005; Founda et al., 2007). A decrease in wind magnitude during eclipse totality was indeed observed in 2019, though results from 2020 are less conclusive. The observed wind magnitudes in 2020 were likely impacted by the passage of a low-level jet associated with strong horizontal water vapor transport which occurred during the campaign. The

differences in variability and overall wind magnitude observed at Toltén and Villarrica (Fig. 4b–4c) are likely explained by the coastal environment at Toltén compared to Villarrica's higher elevation and closer proximity to the Andes Mountains, given other meteorological variables were otherwise similar. Both models simulated a sharp rise in wind magnitude after totality at Villarrica which was far more gradual at Toltén.

Comparisons of vertical air temperature measurements against results from WRF-eclipse show greater model error near the surface during the clear-sky conditions in 2019. In contrast, results from 2020 at both Toltén and Villarrica show the largest model errors occurring aloft, at the 100 hPa pressure level (Fig. 6). Overall, the RMSE and MAE values indicate better agreement between models and observations for the 2020 eclipse (Table 1).

In conclusion, the above results from this unique comparison of surface and profile observations indicate that the WRF-eclipse model is indeed capable of simulating atmospheric responses to an eclipse with reasonable accuracy, however overall accuracy across meteorological variables is shown to be partly influenced by local or regional atmospheric conditions. Beyond applications of this dataset to future atmospheric and modeling studies, additional research is recommended to validate the performance of other WRF-ARW physics schemes since this study does not

attempt to provide a comprehensive assessment of the multiple physics and dynamics options available within WRF-ARW. PBL schemes within WRF-ARW are of particular interest since eclipse-induced atmospheric responses are expected primarily in this lowest region of the atmosphere.

One future research opportunity, the Nationwide Eclipse Ballooning Project (https://eclipse.montana.edu/), will

perform hourly radiosonde measurements from sites across the U.S. during eclipses in 2023 and 2024. This project will result in an abundance of atmospheric profile and surface data for before, during, and after the eclipses which would be highly valuable for further validation studies of the WRF-eclipse model to expand on the results presented here. Additionally, since previous eclipse ballooning campaigns have focused TSEs, the 14 October 2023 annular



eclipse presents an opportunity to perform measurements and subsequent validation studies for an additional type of
solar eclipse.

**Data Availability**

The datasets presented here are publicly available from https://osf.io/894jr/.

**Code Availability**

The algorithms and software used in collection and analyses of the data presented here are publicly available from the
corresponding author upon reasonable request.

**Author Contributions**

The 2020 eclipse campaign was conceived by JF and CS. The methodology was completed by GM and CS. CS drafted
the manuscript and JF, JW, and GM provided reviews and revisional suggestions. Manuscript map created by CS and
figure plots were created by GM.

**Competing Interests**

The authors declare that they have no conflict of interest.

**Acknowledgements**

This research was funded by NSF AGS award #218182 and NASA Montana Space Grant Consortium. We gratefully
acknowledge the support of the following organizations and individuals: Chilean Directorate General of Civil
Aviation; Chilean Ministry of Health; Chilean Ministry of Science, Technology, Knowledge and Innovation; the U.S.
Embassy in Santiago, E. Isaman; Universidad de Santiago, Dr. J. Carrasco, and the Consulate General of Chile.
Students, faculty, and staff of University of Montana, Carroll College, University of Idaho, Oklahoma State
University, and University of Kentucky also contributed greatly to the project's success including data collection and
analysis efforts.

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




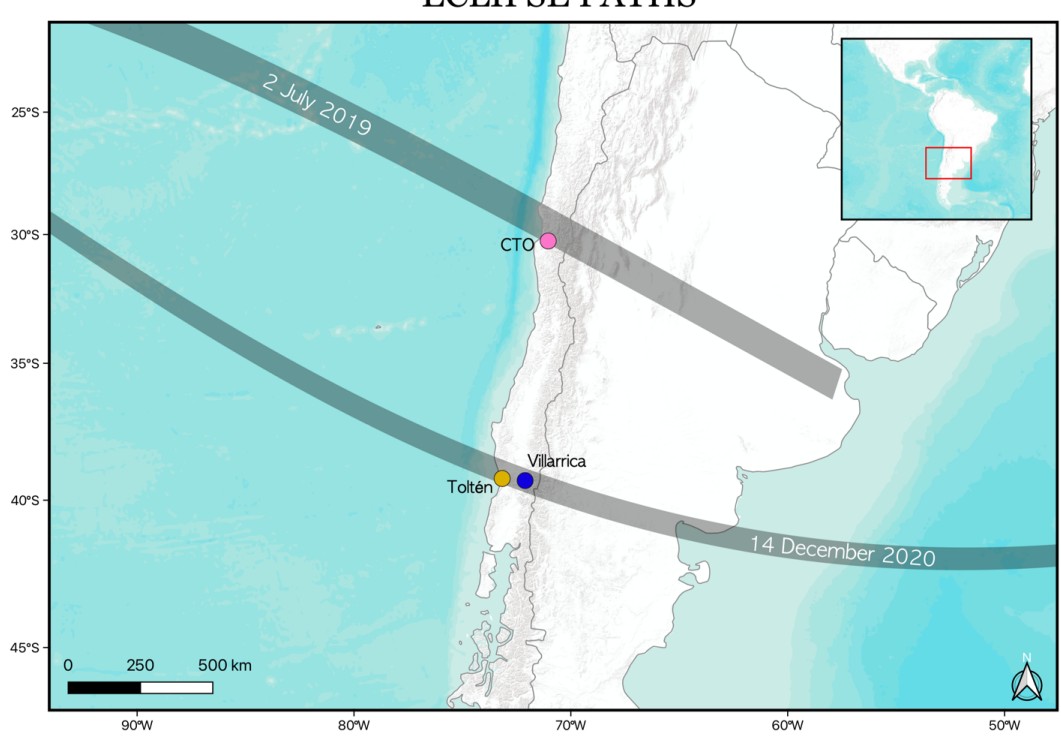

**Figure 1: Map of eclipse totality paths (grey shaded areas) for the 2 July 2019 and 14 December 2020 TSEs. The 2019 launch site at Collowara Tourism Observatory (CTO; pink point) and 2020 launch sites, Toltén (yellow point) and Villarrica (blue point) are shown. Map created by Carl Spangrude using QGIS software ver. 3.22.0. Eclipse paths courtesy of Xavier M. Jubier (xjubier.free.fr/en/site_pages/SolarEclipsesGoogleEarth.html).**






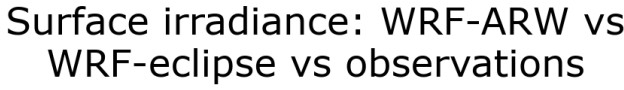

Figure 2: Comparison of irradiance (W m⁻²) observations (solid blue lines) and modeled results at the surface for the 2 July 2019 eclipse (a) and from two sites (Toltén (b); Villarrica (c)) during the 14 December 2020 eclipse. Modeled results include WRF simulations with the eclipse enabled (dashed green lines) and disabled (dashed orange lines). Times of the eclipses are marked by the shaded grey area. Surface measurements from Andacollo, Chile in 2019 concluded at 22:30:00 UTC. Prior to the eclipse, WRF-ARW results are masked by WRF-eclipse results since both were identical until C1.



**Figure 3: Comparison of temperature (°C) observations at 2 m (solid blue lines) and modeled results at 2 m for the 2 July 2019 eclipse (a) and from two sites (Toltén (b); Villarrica (c)) during the 14 December 2020 eclipse. Modeled results include WRF simulations with the eclipse enabled (dashed green lines) and disabled (dashed orange lines). Times of the eclipses are marked by the shaded grey area. Surface measurements from Andacollo, Chile in 2019 concluded at 22:30:00 UTC. Prior to the eclipse, WRF-ARW results are masked by WRF-eclipse results since both were identical until C1.**



Figure 4: Comparison of wind speed (m s⁻¹) observations at 2 m (solid grey lines), observations at 10 m (solid blue lines), and modeled results at 10 m for the 2 July 2019 eclipse (a) and from two sites (Toltén (b); Villarrica (c)) during the 14 December 2020 eclipse. Modeled results include WRF simulations with the eclipse enabled (dashed green lines) and disabled (dashed orange lines). Surface station observations are 3 min averages. Times of the eclipses are marked by the shaded grey area. Surface measurements from Andacollo, Chile in 2019 concluded at 22:30:00 UTC. Prior to the eclipse, WRF-ARW results are masked by WRF-eclipse results since both were identical until C1.



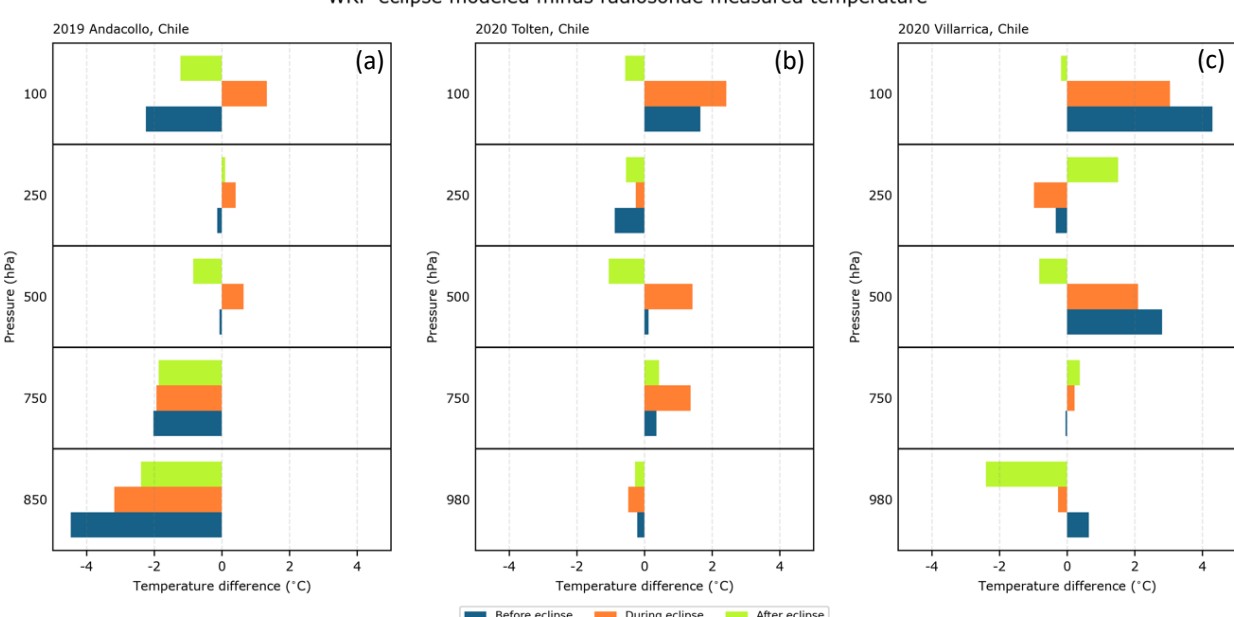

**Figure 5: Differences between WRF-eclipse results and profile observations of air temperature (°C) before (blue), during (orange), and after (green) the eclipses of 2019 (a) and 2020 (b–c). For Andacollo (a), before eclipse data are from flight 21, during eclipse data are from flight 23, and after eclipse data are from flight 25 (launched 2 July 2019 at 17:23:00 UTC, 19:58:50 UTC, and 22:35:39 UTC, respectively. Eclipse totality occurred at 20:39:17 UTC.). For Toltén (b), before eclipse data are from flight 21, during eclipse data are from flight 24, and after eclipse data are from flight 27 (launched 14 December 2020 at 12:01:19 UTC, 14:59:02 UTC, and 17:30:13 UTC, respectively. Eclipse totality occurred at 16:01:45 UTC.). For Villarrica (c), before eclipse data are from flight 21, during eclipse data are from flight 24, and after eclipse data are from flight 27 (launched 14 December 2020 at 12:02:21 UTC, 15:02:03 UTC, and 18:02:00 UTC, respectively. Eclipse totality occurred at 16:04:12 UTC.). Negative values indicate underestimation of air temperature by the model relative to observations; positive values indicate overestimation by the model. Differences are calculated from pressure levels (hPa) 980, 850, 750, 500, 250, and 100 for profile data and at the closest corresponding model grid altitudes. For Andacollo, 850 hPa is substituted for the 980 hPa level given the high elevation of the Andacollo launch site.**



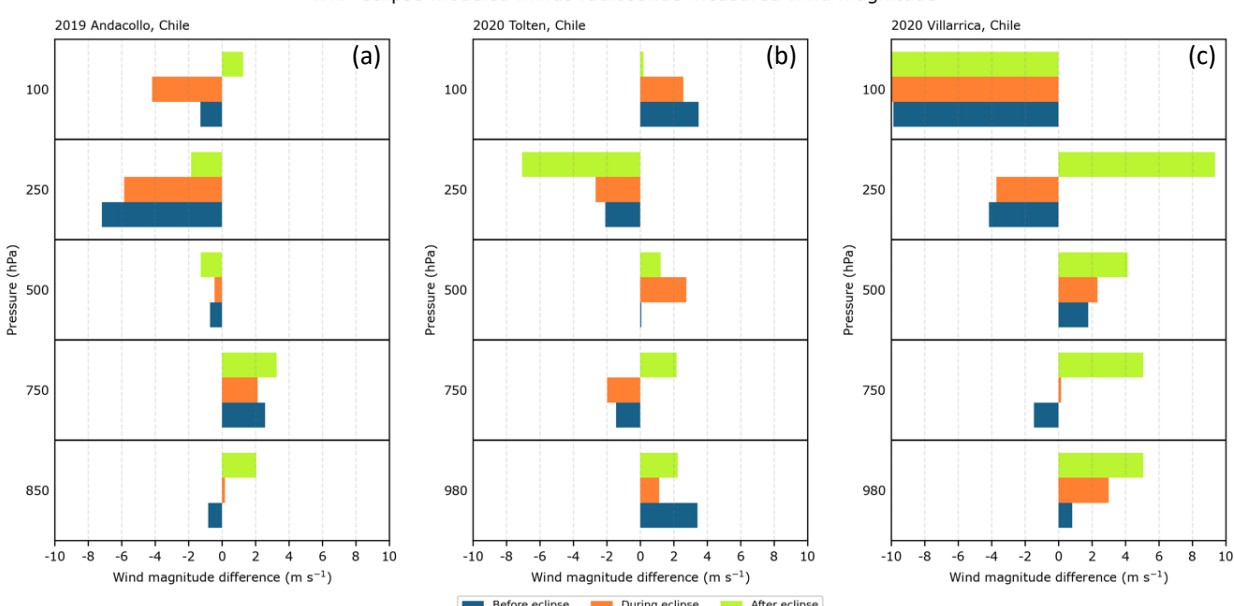

**Figure 6: Differences between WRF-eclipse results and profile observations of wind magnitude (m s$^{-1}$) before (blue), during (orange), and after (green) the eclipses of 2019 (a) and 2020 (b–c). For Andacollo (a), before eclipse data are from flight 21, during eclipse data are from flight 23, and after eclipse data are from flight 25 (launched 2 July 2019 at 17:23:00 UTC, 19:58:50 UTC, and 22:35:39 UTC, respectively. Eclipse totality occurred at 20:39:17 UTC.). For Toltén (b), before eclipse data are from flight 21, during eclipse data are from flight 24, and after eclipse data are from flight 27 (launched 14 December 2020 at 12:01:19 UTC, 14:59:02 UTC, and 17:30:13 UTC, respectively. Eclipse totality occurred at 16:01:45 UTC.). For Villarrica (c), before eclipse data are from flight 21, during eclipse data are from flight 24, and after eclipse data are from flight 27 (launched 14 December 2020 at 12:02:21 UTC, 15:02:03 UTC, and 18:02:00 UTC, respectively. Eclipse totality occurred at 16:04:12 UTC.). Negative values indicate underestimation of wind magnitude by the model relative to observations; positive values indicate overestimation by the model. Differences are calculated from pressure levels (hPa) 980, 850, 750, 500, 250, and 100 for profile data and at the closest corresponding model grid altitudes. For Andacollo, 850 hPa is substituted for the 980 hPa level given the high elevation of the Andacollo launch site.**



730    **Table 1: RMSE and MAE values for upper air data and WRF-eclipse model comparison**

| Temperature Before Eclipse | | | |
|---|---|---|---|
| | 2019 Andacollo, Chile | 2020 Toltén, Chile | 2020 Villarrica, Chile |
| MAE | 1.8 °C | 0.6 °C | 1.6 °C |
| RMSE | 2.4 °C | 0.8 °C | 2.3 °C |
| Temperature During Eclipse | | | |
| | 2019 Andacollo, Chile | 2020 Toltén, Chile | 2020 Villarrica, Chile |
| MAE | 1.5 °C | 1.2 °C | 1.3 °C |
| RMSE | 1.8 °C | 1.4 °C | 1.7 °C |
| Temperature After Eclipse | | | |
| | 2019 Andacollo, Chile | 2020 Toltén, Chile | 2020 Villarrica, Chile |
| MAE | 1.2 °C | 0.6 °C | 1.6 °C |
| RMSE | 1.5 °C | 0.6 °C | 1.8 °C |
| Wind Speed Before Eclipse | | | |
| | 2019 Andacollo, Chile | 2020 Toltén, Chile | 2020 Villarrica, Chile |
| MAE | 2.5 m s$^{-1}$ | 2.1 m s$^{-1}$ | 3.6 m s$^{-1}$ |
| RMSE | 3.5 m s$^{-1}$ | 2.5 m s$^{-1}$ | 4.9 m s$^{-1}$ |
| Wind Speed During Eclipse | | | |
| | 2019 Andacollo, Chile | 2020 Toltén, Chile | 2020 Villarrica, Chile |
| MAE | 2.5 m s$^{-1}$ | 2.2 m s$^{-1}$ | 5.4 m s$^{-1}$ |
| RMSE | 3.3 m s$^{-1}$ | 2.3 m s$^{-1}$ | 8.2 m s$^{-1}$ |
| Wind Speed After Eclipse | | | |
| | 2019 Andacollo, Chile | 2020 Toltén, Chile | 2020 Villarrica, Chile |
| MAE | 1.9 m s$^{-1}$ | 2.6 m s$^{-1}$ | 5.9 m s$^{-1}$ |
| RMSE | 2.1 m s$^{-1}$ | 3.5 m s$^{-1}$ | 6.2 m s$^{-1}$ |