# Peer review of "Validation of the WRF-ARW Eclipse Model with Measurements from the 2019 & 2020 Total Solar Eclipses"

_EGUsphere, 2023_

## Author Response (AR1)

Point-by-point response to referee comments
*Spangrude et al. 2023*

Anonymous Referee #1

- Referee comment RC1:

This paper presents model simulations including an eclipse parameterization during two South American eclipses at 3 sites. The field campaigns were clearly very extensive with multiple radiosonde launches aimed at looking at stratospheric gravity waves. However, the model comparison part of the paper really only validates the surface eclipse simulation which works quite well. The upper air treatment is quite superficial and model validation only uses the eclipse version to get mean errors in selected layers around the eclipse time. My main recommendation is to try to improve the upper-air part of the paper. While it may be difficult to detect gravity wave activity in the radiosonde data, and I am sure that is a separate paper, the model with and without the eclipse may have shown a signal worth presenting. Can the authors look at differences between the two simulations over an area to see if any systematic signal is detected in temperature, winds, or vertical motion. Such signals could be used in guidance of what to look for in radiosonde data. This would address a clear gap between the field program goals and the model simulations presented.

- Author Response AC1:

The authors agree that the extent of upper air model validation is limited. The upper air validation presented is not intended to be a comprehensive model analysis, but rather to provide preliminary comparisons between model results and observations not previously described in the literature, in line with the stated goals of the manuscript. The datasets analyzed and model configurations used here are shared publicly to enable further validation by those interested in performing more in-depth upper air analyses.

Gravity wave analysis using radiosonde data was performed for the 2019 campaign, described in detail by Colligan et al. 2020, and cited in this work. Given the present paper's focus on model validation and not on gravity waves, in depth gravity wave analysis using model data is beyond the scope of this work, though may be a focus of future manuscripts authored by the 2020 campaign teams and collaborators. Accordingly, gaps between results presented here and goals of the field campaigns are envisioned to be addressed in subsequent manuscripts.

- Referee comment RC2:

I was thinking of just seeing if gravity wave signals were detectable in the differences between your simulations. It seems in the scope of the paper just to show a model

difference plot of some kind that illustrates these waves. Given the scale of the shadow, these would presumably be quite long and may have inertial rotation characteristics.

- Author Response AC2:

This feedback is appreciated. The authors plan to incorporate such a model difference plot, along with associated results and conclusions, in a revised version of the manuscript. This will address the gap between the campaign goals and the model results presented as well as expand the upper air analysis portion as originally suggested.

- Changes in revised manuscript:

Line #86: Added additional clarity to the stated goals of the manuscript.
Lines #326-327 & #391-393: Added acknowledgement of the preliminary nature of the upper air analysis performed.
Lines #345-357: Added descriptions of the results from the recommended additional analysis comparing eclipse and no-eclipse model simulations for 2019 and 2020 to illustrate potential model differences with regard to atmospheric gravity wave signals.
Lines #403-407: Added discussion of the above results to the conclusions section.
Lines #790-878: Added two additional figures showing model differences in potential temperature and vertical velocity for eclipse and no-eclipse simulations from the 2019 and 2020 TSEs to identify possible eclipse-related signals in atmospheric gravity waves.

Anonymous Referee #2

- Referee comment RC3:

This paper evaluates the WRF-ARW eclipse model again radiosonde observations during an eclipse in South America. The authors accomplish their objectives to (1) compare measurements from two different field campaigns of two different eclipses and (2) present preliminary results evaluating the performance of the WRF eclipse model in simulating the response to the eclipse.

The availability of data from hourly radiosonde launches is particularly unique to this study of eclipses as shown in Colligan et al.; although, the present form of this paper doesn't provide a very extensive analysis of the profile observations compared to WRF simulations. It would be informative to show a comparison of the full radiosonde profiles next to profiles of the WRF and WRF-eclipse simulations rather than a selection of statistics presently shown in the last two figures.

One other suggestion is to provide the total number of data points considered for each statistic in Table 1 and confidence intervals where appropriate since the authors use words like "high accuracy" and "significant temperature decrease."

- Author Response AC3:

The authors appreciate this feedback. We agree that an extensive analysis is not provided. This is intentional given the manuscript's stated goals.

As the paper states: "This study presents basic WRF-eclipse model validation of not only surface variables but also – for the first time – preliminary validation using profile comparisons." This paper is not intended to provide an in-depth evaluation of all the profiles collected, but rather is a just a preliminary evaluation of the WRF model performance with more extensive analysis to follow.

We have chosen to represent uncertainties through MAE and RMSE instead of confidence intervals as we are not endeavoring to provide a full assessment of model performance, but only the error with our small sample size. A more extensive analysis with a larger sample size can follow this paper. The number of data points considered are limited to those visually represented in Figures 5 and 6, thus warranting the further analysis we suggest. A revised version of the manuscript will adjust the language used to reflect the authors acknowledgment of the limitations of the small sample size of the present analysis.

- Changes in revised manuscript:

Lines #326-327 & #391-393: Added acknowledgement of the preliminary nature of the upper air analysis performed and limitations of the relatively small sample size in the present analysis.

Lines #292, #362, #369, #388, & #390: Adjusted language used to reflect the preliminary nature of the analyses and the limited representativeness of results based on the relatively small sample sizes considered.